# Longevity and marginal bone loss of narrow-diameter implants supporting single crowns: A systematic review

**Lucas Henrique Telles[1], Fernando Freitas Portella****[1,2], Elken Gomes Rivaldo[1]\***

**1** Programa de Pós-Graduação em Odontologia, Universidade Luterana do Brasil, Canoas, RS, Brazil,
**2** Universidade Feevale, Novo Hamburgo, RS, Brazil

\* elkenrivaldo@gmail.com

**Data Availability Statement:** All relevant data are within the manuscript and its Supporting Information files.

## Abstract

### Purpose

To compare the longevity and marginal bone loss of narrow-diameter (≤3.3-mm) versus standard-diameter implants supporting single crowns.

### Material and methods

The MEDLINE (via PubMed), Scopus, and SciELO databases were searched for relevant publications. In addition, the scientific references provided by each of the implant companies that appeared in the search were reviewed. Intervention studies comparing longevity and bone loss between narrow-diameter and standard-diameter implants were included.

### Results

The search was limited to *in vivo* studies in humans. The query returned 1931 results, of which 4 met the inclusion criteria. The implant success rate ranged from 93.8% to 100% over a maximum follow-up of 3 years, with no difference between narrow- and standard-diameter implants. Meta-analysis of all included studies showed greater bone loss in narrow-diameter implants as compared with standard ones; however, when analysis was restricted to randomized trials, no such difference was present.

### Conclusion

The meta-analysis showed no difference in longevity between narrow implants and standard implants when supporting single crowns. However, narrow-diameter implants may be associated with greater marginal bone loss. These findings should be regarded cautiously due to the short follow-up duration and methodological heterogeneity of the primary studies.

**Funding:** The funders had no role in study design, data collection and analysis, decision to publish, or preparation of the manuscript.

**Competing interests:** The authors have declared that no competing interests exist.

## Introduction

Narrow-diameter implants (<3.3 mm) were designed for edentulous sites with reduced mesio-distal space, such as mandibular central and lateral incisors, as well as maxillary lateral incisors. The current literature supports their extended use in other clinical situations as well, such as narrow alveolar ridges, to prevent preoperative or intraoperative bone reconstruction [1,2] (which increase treatment cost and duration [3,4]) and to reduce postoperative morbidity. However, commercially pure titanium (cpTi) has limited mechanical strength, which led to the hypothesis that the use of narrow-diameter implants could pose a risk of fracture of the implant body [5,6]. Advances in biomaterials led to the development of alloys with increasing fracture and fatigue resistance, such as an alloy of titanium and zirconium (83–87% titanium added to 13–17% zirconium [7]). Biomechanical assays of this alloy in experimental models demonstrated increased resistance to the stresses generated by occlusal loading compared to cpTi [3], even at smaller-than-conventional diameters.

Observational studies have shown that narrow-diameter implants have comparable longevity to standard implants [8,9]. In addition, when the longevity of narrow-diameter and standard implants was compared, that of implants with a diameter of 3.0–3.25 mm was found not to differ from that of standard-diameter implants [4,10]. Nonetheless, this finding is based predominantly on observational studies, and does not distinguish between different types of prosthetics supported by these implants. Thus, it does not answer the question of whether practitioners can choose to place narrow-diameter implants instead of conventional ones for the rehabilitation of single missing teeth. Furthermore, there is no consensus on the influence of implant diameter on marginal bone loss. The present review focused on a problem not previously addressed in the literature: narrow-diameter implants (those implants with a diameter of 3.3 mm or less) supporting only single crowns. Within this context, the present systematic review was designed to compare longevity and marginal bone loss between narrow-diameter and standard-diameter implants supporting single crowns.

## Methods

The present systematic review was registered in the PROSPERO platform (CRD42018117261) and reported in accordance with the PRISMA statement [11]. The guiding research question for this study was "Is there a difference in longevity and marginal bone loss between narrow-diameter and standard-diameter implants when supporting single crowns in healthy patients?". In PICO format, this question may be phrased as follows:

P–healthy patients;

I–implants supporting single crowns;

C–narrow-diameter versus standard-diameter implants;

O–longevity of implant and marginal bone loss.

### Eligibility criteria

Publications that met the following eligibility criteria were included:

a. intervention studies in humans which evaluated the longevity of implants retaining single crowns; observational prospective studies were not included.

b. with at least one group of patients which received narrow-diameter implants and one group which received standard-diameter implants, with a minimum of 10 participants in each group;

c. which reported implant longevity over a follow-up period of at least 1 year.

There were no restrictions on language or date of publication.

## Sources and search strategy

The MEDLINE (via PubMed), Scopus, and SciELO databases were searched through December 2018. The search strategy for each database is described below.

PubMed: (("dental implant" OR "dental implants" OR "narrow implant" OR "narrow implants" OR "narrow-diameter implant" OR "narrow-diameter implants")) AND (longevity OR survival OR "follow-up" OR success)) AND ("randomized controlled trial" OR RCT OR "clinical study" OR trial OR "clinical research" OR "longitudinal study" OR "cohort study" OR practice-based);

Scopus: (TITLE-ABS-KEY ("dental implant" OR "dental implants" OR "narrow implant" OR "narrow implants" OR "narrow-diameter implant" OR "narrow-diameter implants") AND TITLE-ABS-KEY (longevity OR survival OR "follow-up" OR success) AND TITLE-ABS-KEY ("randomized controlled trial" OR rct OR "clinical study" OR trial OR "clinical research" OR "longitudinal study" OR "cohort study" OR practice-based));

SciELO: "dental implants";

In addition to the aforementioned electronic databases, the scientific literature recommended by implant manufacturers as a reference on their products was also reviewed. The following manufacturers were included: Straumann (Switzerland), Nobel Biocare (Switzerland), Biohorizons (USA), Z-systems (Germany), Dentsply (USA), 3i (USA), and Dentatus (Sweden).

## Study selection and data collection process

The article lists retrieved by each search strategy were unified and duplicate entries removed. For study selection, two investigators (LHT and EGR) evaluated the titles and abstracts separately. Those considered ineligible by both reviewers were excluded outright; those considered ineligible by one reviewer but eligible by the other were retained for full-text reading.

All studies not excluded were read in full by two investigators working together (LHT and FFP), who then selected those that fully met the eligibility criteria and performed data extraction. At this time, the authors also performed the risk of bias analysis. Patient-related and procedure-related data were collected, as were the outcomes of interest (longevity and marginal bone loss).

Longevity was determined by the percentage of implants considered successful at the latest recorded time point of clinical evaluation. Implants were considered successful when they were osseointegrated and remained functional. Marginal bone loss data were collected when assessed by radiographic examination; mean values for implant sites were calculated for analysis. To perform sensitivity analysis, the number of implants that were not followed in each experimental group was recorded, and longevity was then recalculated. When more than one publication was derived from a single study, with different follow-up periods for the same sample, data from all publications referring to the same study were pooled and the publication with the longest duration of follow-up was recorded in the table.

## Risk of bias in individual studies

Studies were assessed for risk of bias as recommended in the *Cochrane Handbook of Systematic Reviews of Interventions*. The possibility of conflict of interest with implant manufacturers was also considered and recorded.

## Data synthesis and analysis

Implant longevity was recorded as the number of successful cases followed in relation to the total number of cases in each group. Marginal bone loss around the implant head was extracted in mm for each of the experimental groups.

The outcomes of the included articles were pooled for meta-analysis to determine the overall estimated effects of each outcome based on a random effects model. For analysis of longevity, the Mantel-Haenszel method was applied to calculate the risk ratio for success, while for marginal bone loss, the overall mean difference was calculated. Statistical analysis was performed in RevMan 5.3 software (The Cochrane Collaboration). Significance was accepted at the 5% level.

# Results

## Study selection

Fig 1 provides a flow diagram of study selection. The database search (PubMed, Scopus, and SciELO) retrieved 6,402 articles. A hand search of the reference lists provided by manufacturers yielded a further 1,677. After exclusion of duplicate entries, 1,931 papers remained. Of

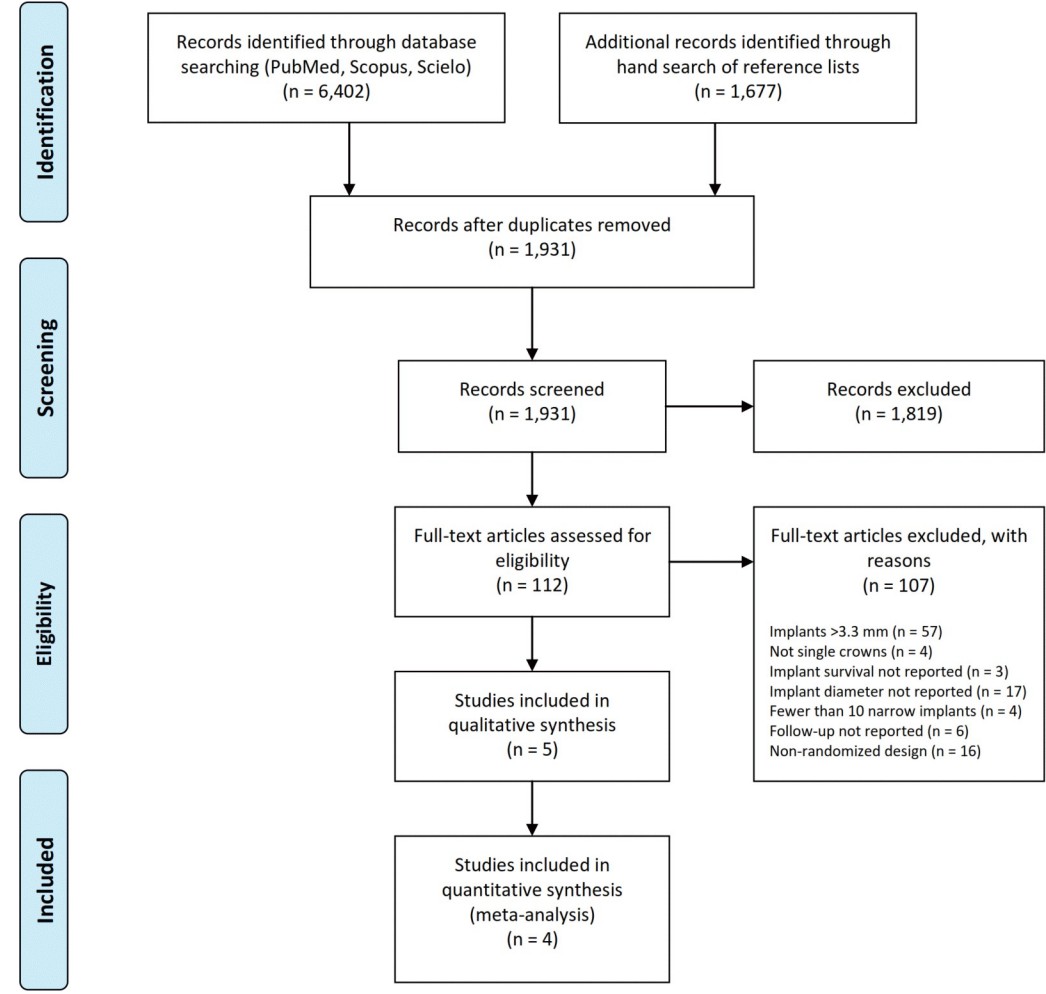

**Fig 1. Flow diagram of study selection.**

**Table 1. Characteristics of the studies and detailed description of comparison groups.**

| Study | Andersen (2001) [4] | | Santing (2013) [14] | | Ioannidis (2015) [8,12] | | de Souza (2018) [13] | |
|---|---|---|---|---|---|---|---|---|
| | **Narrow** | **Standard** | **Narrow** | **Standard** | **Narrow** | **Standard** | **Narrow** | **Standard** |
| **No. patients** | 28 | 27 | 12 | 48 | 20 | 20 | 22 | 22 |
| **Mean age (years)** | 23.2 | 23.0 | 36.9 | | Not reported | | 53.4 | |
| **No. implants** | 32 | 28 | 12 | 48 | 20 | 20 | 22 | 22 |
| **Diameter** | 3.25 | 3.75 | 3.3 | 4.1 | 3.3 | 4.1 | 3.3 | 4.1 |
| **Length** | 13 and 15 | 13 and 15 | 12 and 14 | 12 and 14 | ≥8 | ≥8 | 6 to 12 | 8 to 10 |
| **Implant site** | Anterior maxilla | | Anterior maxilla | | Anterior or posterior, maxilla or mandible | | Posterior, maxilla or mandible | |
| **Manufacturer** | 3i (2-piece) | | Straumann (2-piece) | | Straumann (2-piece) | | Straumann (2-piece) | |
| **Platform type** | Platform matching | | Platform switching | | Platform switching | | Platform matching | |
| **Apical-coronal implant position** | Bone level | | Bone level | | Bone level | | Tissue level | |
| **Implant material** | Titanium | | Titanium- Zirconium /Titanium | | Titanium- Zirconium /Titanium | | Titanium- Zirconium /Titanium | |
| **Surface treatment** | No | | Yes | | Yes | | Yes | |
| **Time to loading** | 6 months | | 3 months | | 3 months | | Not reported | |
| **Duration of follow-up** | 3 years | | 1.5 year | | 3 years | | 3 years | |
| **Success rate at latest follow-up** | 93.8% | 100% | 100% | 100% | 100% | 100% | 95.4% | 100% |
| **Outcome of followed implants** | 27 successful, 3 lost to follow-up | 26 successful, 2 lost to follow-up | All 12 successful | All 48 successful | 17 successful, 3 lost to follow-up | 15 successful, 5 lost to follow-up | 19 successful, 2 lost to follow-up, 1 failed | 20 successful, 2 lost to follow-up |
| **Radiographic marginal bone loss (mm)** | 0.52±0.01 | 0.40±0.16 | Not reported | | 0.40±0.93 | 0.31±0.59 | 0.58±0.39 | 0.53±0.46 |

these, 1,819 were excluded after analysis of titles and abstracts. Thus, 112 articles were read in full. None of these was found only on manufacturers' reference lists. After full-text reading, 107 papers were removed for failure to meet the eligibility criteria. Of the 5 eligible articles remaining, two were from the same study, but with different follow-up times [8,12]). The 4 remaining articles were included in the meta-analysis (Table 1).

## Study characteristics

De Souza et al [13]. conducted a split-mouth randomized clinical trial in which patients received Tissue Level implants (Straumann, Switzerland) with a diameter of 3.3 mm at one site and 4.1 mm in another. Each patient received at least one implant of each diameter in the maxilla or mandible, for a total of 44 implants placed (22 narrow and 22 standard). Periapical radiographs were performed on the day of placement and at 1-year and 3-year follow-up. The authors found no significant differences in marginal bone loss at any of the three follow-up time points (immediate, 1 year, and 3 years). After 3 years of follow-up, the success rate was 100% for standard-diameter implants and 95.4% for narrow-diameter implants. One implant presented a pocket depth >5 mm, with bleeding on probing and suppuration. Nevertheless, the implant remained in position and functional, for a 100% survival rate in the narrow-diameter group.

Santing et al [14] carried out a prospective intervention study using Bone Level implants (Straumann, Switzerland) for single-tooth replacement in the maxilla. Sixty patients participated in the study, with each receiving a 3.3-mm (narrow) or 4.1-mm (standard) implant.

Overall, 12 narrow and 48 standard implants were placed. The authors compared bone loss on periapical radiographs between implants placed in augmented sites and those placed in native bone. There was no significant difference between groups, with a 100% survival rate at a maximum follow-up time of 18 months.

In a randomized clinical trial, Ioannidis et al [8,12] sought to compare loss of marginal bone level with 3.3-mm versus 4.1-mm implants. Implants in the experiment group (n = 20) were made from a Ti-Zr alloy, while those in the control group (n = 20) used conventional titanium alloys. The implants were placed randomly at sites requiring single tooth replacement, at any location in the maxilla or mandible, and were followed for up to 3 years. At 1-year follow-up, all evaluated implants were in place and stable; two participants in the control group were subsequently lost to follow-up. At the end of follow-up, 32 of the 40 included patients were examined (15 control, 17 experiment). The other 8 patients were lost to follow-up. According to the authors, there were no implant failures, giving a 100% survival rate for both groups over 3 years (among the implants examined at final follow-up).

A clinical trial by Andersen et al.[4] included 55 patients requiring single-tooth replacements in the anterior region of the maxilla. Of these, 27 received 28 standard implants (diameter 3.75 mm), and 28 patients received 32 narrow implants (diameter 3.25 mm). Two implants in the narrow-diameter group were lost (93.8% survival rate), versus none in the standard group (100%).

## Synthesis of results

A meta-analysis considering those intervention studies (Fig 2) that allowed comparison of longevity between narrow and standard implants showed no difference between them (RR 0.98 [0.90, 1.07]). A sensitivity analysis including only the RCT (Fig 3) showed a similar risk ratio for success among distinct diameters, with a slight increase in the 95% confidence interval (RR 1.00 [0.84, 1.20]).

Narrow implants were associated with greater marginal bone loss, with a mean 0.12 (0.06 to 0.18) mm of additional loss compared to standard implants (Fig 4). This difference did not remain when only RCTs were included in the meta-analysis (Fig 5), the mean difference in bone loss then being 0.06 (-0.18 to 0.30) mm.

## Risk of bias

Fig 6 summarizes the analysis of risk of bias in the included studies. None of the studies included all of the quality categories evaluated; blinding of assessors and patients was most often neglected. Three of the four studies had some form of relationship with the industry. Funnel plots analyses showed a low risk of publication bias, while the $I^2$ value of 0% was consistent with absence of statistical heterogeneity among studies, both for longevity and for marginal bone loss.

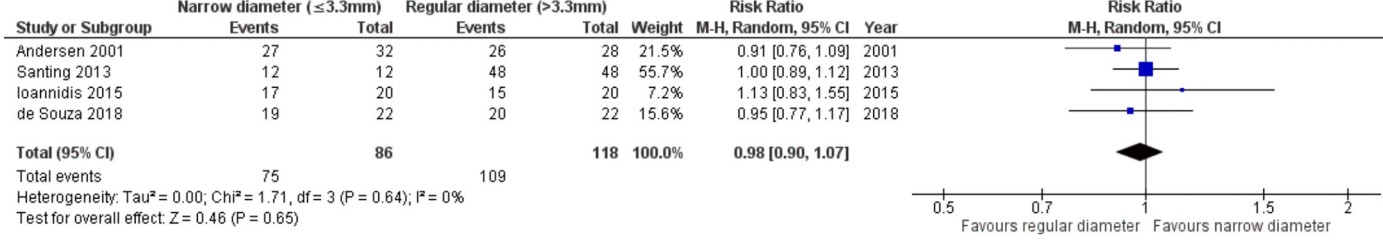

**Fig 2. Forest plot of all intervention studies comparing the longevity of narrow and standard implants.**

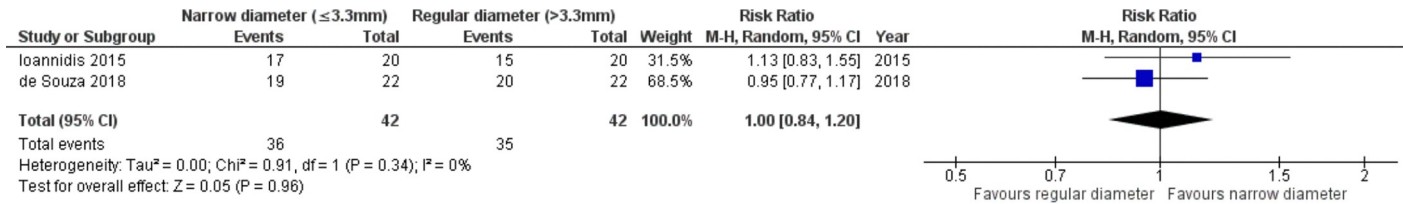

**Fig 3. Forest plot of randomized clinical trials comparing the longevity of narrow and standard implants.**

## Discussion

The present literature review and meta-analysis was designed to help dental practitioners decide which implant diameter to choose in settings that would allow placement of both narrow-diameter and standard implants to support single crowns. Despite the short follow-up period of 3 years or less, longevity was shown not to differ between diameters. However, narrow-diameter implants may be associated with greater marginal bone loss—0.12 (0.06–0.18) mm greater at 3-year follow-up. It bears stressing that the evaluation of implant success goes beyond longevity. Several extrinsic and intrinsic variables may be implicated in peri-implant marginal bone loss. Factors related to the implant itself include its geometry, dimensions and interface between the prosthetic abutment and the implant, the three-dimensional position of the implant, and its angle. Patient-related factors include the quantity and quality of hard and soft tissue surrounding the implant, which are involved in maintaining the biological distances between tooth and implant and from implant to implant [1].

The amount of bone tissue surrounding a narrow-diameter implant was likely to be smaller than when placing a standard implant in the non-randomized studies included in this systematic review. This would justify the greater marginal bone loss with narrow-diameter implants. A randomized clinical trial comparing the amount of peri-implant bone through postoperative CBCT could provide further evidence to support or refute this assumption. In addition to implant diameter, the type of platform and the position of the implant-abutment junction [15] can also influence marginal bone loss. A systematic review comparing marginal bone loss with platform-switching versus platform-matching implants, which pooled the outcomes of 15 publications with data from 642 patients followed up for 1 year, suggested greater bone loss with platform matching [16]. Among the three studies included in the analysis of marginal bone loss in this review, Andersen [4] reported higher values for narrow implants, which accounted for the statistical difference observed in the meta-analysis that included all eligible intervention studies. A stratified analysis, evaluating the joint effect of implant diameter and of different implant platforms and apicocoronal positions, is necessary for a better understanding of the factors associated with marginal bone loss.

The included studies reported on implants placed in different regions of the oral cavity. Santing et al. [14] reported on implants placed in the anterior and first premolar regions, while

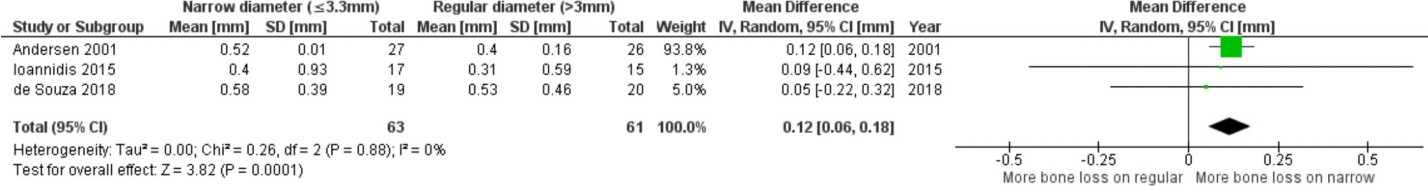

**Fig 4. Forest plot of all intervention studies comparing marginal bone loss between narrow and standard implants.**

| Study or Subgroup | Narrow diameter (≤3.3mm) | | | Regular diameter (>3mm) | | | Weight | Mean Difference IV, Random, 95% CI [mm] | Year |
|---|---|---|---|---|---|---|---|---|---|
| | Mean [mm] | SD [mm] | Total | Mean [mm] | SD [mm] | Total | | | |
| Ioannidis 2015 | 0.4 | 0.93 | 17 | 0.31 | 0.59 | 15 | 20.1% | 0.09 [-0.44, 0.62] | 2015 |
| de Souza 2018 | 0.58 | 0.39 | 19 | 0.53 | 0.46 | 20 | 79.9% | 0.05 [-0.22, 0.32] | 2018 |
| Total (95% CI) | | | 36 | | | 35 | 100.0% | 0.06 [-0.18, 0.30] | |

Heterogeneity: Tau² = 0.00; Chi² = 0.02, df = 1 (P = 0.90); I² = 0%
Test for overall effect: Z = 0.48 (P = 0.63)

More bone loss on regular    More bone loss on narrow

**Fig 5. Forest plot of randomized clinical trials comparing marginal bone loss between narrow and standard implants.**

de Souza et al. [13] placed implants only in the posterior region. Ioannidis et al [8,12] placed implants in the anterior and premolar area, and Andersen et al. [4], in the anterior maxilla. This variability hindered analysis of the potential impact of implant location. Another aspect

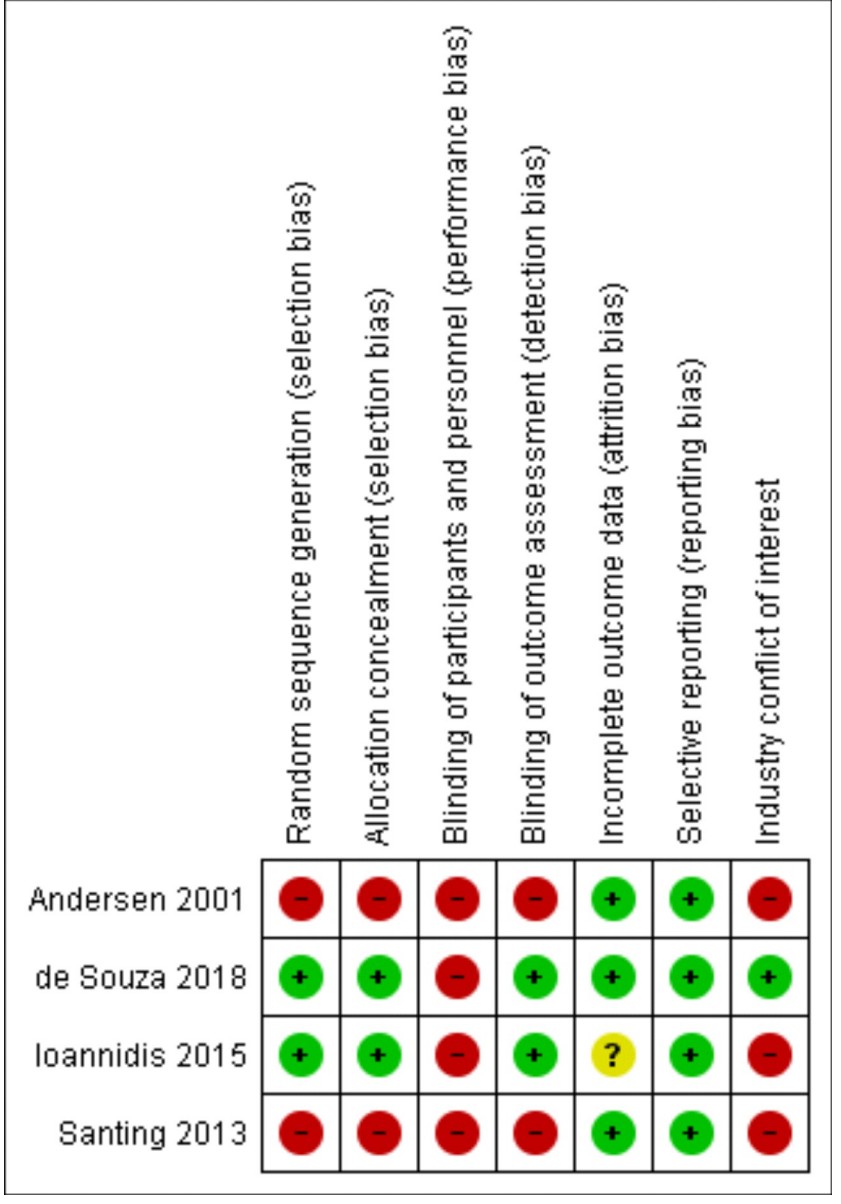

**Fig 6. Risk of bias of included studies.**

to be considered in the data analysis is the reason for tooth loss. Of the articles included in this review, only Santing et al. reported this information; the most frequent cause in their series was tooth fracture (n = 29), followed by endodontic failure (n = 19), root resorption (n = 5), periodontal disease (n = 4) and agenesis (n = 3). Andersen et al. [4] performed immediate pro-visionalization in the narrow implants. Although there is concern about the incidence of masticatory forces on the implants immediately after placement, clinical maneuvers to minimize osseointegration failure and marginal bone loss may be employed regardless of implant diameter [17]. These unique characteristics of the different included studies should be considered when analyzing the results of this meta-analysis.

Longevity or success can be assessed from several aspects [18]. In this review, the criteria for implant longevity were osseointegration and preserved function at clinical evaluation. Implant failures reported in primary studies occur most often in the earliest stage of follow-up (first 6 months), mainly during the process of osseointegration. After this initial period, there were practically no implant failures. However, the narrow implants were associated with greater marginal bone loss, which clinically, in the long term, may represent a concern to the clinician, considering the possibility of cosmetic issues such as retraction of the peri-implant mucosa due to decreased bone support, as well as facilitating the development of peri-implantitis [19].

A previous systematic review with meta-analysis published addressing the comparison narrow-diameter (≤3.3-mm) versus standard-diameter implants [20,21]. In general, these studies showed no difference in longevity and marginal bone loss between diameters. These previous studies grouped implants without distinction as to the type of prosthesis supported. Their survival data were also mixed, referring to narrow implants rehabilitating fully edentulous arches (i.e., supporting removable or fixed complete dentures), as well as partially edentulous ones (supporting removable partial dentures, fixed partial dentures, and single crowns). Furthermore, observational studies were grouped with interventional studies. The decision to limit this systematic review to implants of 3.3 mm or narrower diameter and supporting single crowns only aimed to present clinicians with data on the survival of these biomaterials, which provide advantages such as less need for alveolar ridge augmentation prior to implant placement, less need for orthodontic movement to recover reduced prosthetic spaces, and improved aesthetic outcomes in the anterior zone [22] enhancing the quality of peri-implant tissues. Another, more recent review [23] also focused on studies of narrow implants, comparing these with standard-diameter implants, and including subgroup analyses stratified by diameter (<3 mm, <3.0–3.25 mm, and 3.3–3.5 mm). This review found that the longevity of implants less than 3.0 mm in diameter was shorter than that of standard implants. However, this review also failed to discriminate between implants as to the type of prosthesis supported (single or multiple, fixed or removable). Non-splinted implants are subject to greater overload than those splinted by prosthetic structures [24]. Furthermore, we present a meta-analysis of marginal bone loss.

We chose to conduct our analysis considering implants lost to follow-up as failures and then recalculate longevity, in order to complete a sensitivity analysis, seeking to attenuate any reporting bias. However, considering the longevity reported by the authors of the primary studies, without sensitivity analysis, the effect size measures of the meta-analyses would remain similar, with a risk ratio for success of narrow implants of 0.98 [0.92, 1.03]. Such sensitivity analyses are of the utmost importance when studying treatment longevity, since patient losses to follow-up can have a significant impact on estimated longevity. Considering the success rate of narrow-diameter implants in observational studies, a sensitivity analysis considering losses to follow-up might reduce success rates from nearly 100% to less than 90% [4,12,13,25].

The studies were heterogeneous, with differences in design, implant manufacturers, implant sites (anterior versus posterior, maxilla versus mandible), implant diameters compared, and time to implant loading. It is plausible that some difference in implant longevity would exist as a result of certain variables; however, we were unable to isolate these. In addition, quality evaluation of the studies revealed potential risk of bias, due to issues such as inadequate randomization, absence of allocation concealment or assessor blinding, and industry involvement in all included studies. Due to the inclusion criteria adopted, which restricted the review to intervention studies (not necessarily randomized clinical trials), the absence of random allocation was expected. The selection bias caused by the inclusion of non-randomized studies, besides the methodological heterogeneity of the studies, means that caution is warranted when translating the results of this review into clinical practice. One example is analysis of marginal bone loss, the results of which differed when all studies were included and when analysis was restricted to RCTs.

We decided to include non-randomized intervention studies because of the nature of the research question. It was expected that, in primary studies, allocation of patients into the narrow-diameter and standard-diameter groups would occur according to the characteristics of the enrolled patients, such as the space available for prosthetic rehabilitation, and not by randomization. Thus, the presence of non-randomized comparative studies was expected, and their inclusion allowed us to better answer our research question in this poorly studied context. Nevertheless, on meta-analysis, no evidence of statistical heterogeneity was observed ($I^2 = 0.00$), thus allowing the data to be compiled for analysis.

Incorrect three-dimensional positioning of the implant in the alveolar ridge can lead to physiological bone loss. The coronal portion of bone tends to undergo resorption if the implant is placed too close to neighboring teeth, to other implants, or to a very thin bony wall, as is usually the case with the buccal bone plate [16]. This would justify the practitioner's decision to place an implant with a narrower diameter, in an attempt to maintain a thicker portion of bone tissue around the implant or avoid undue proximity to neighboring teeth or appliances, in order to minimize or mitigate unnecessary resorption.

The meta-analysis showed no difference in longevity between narrow implants and standard implants when supporting single crowns. However, narrow-diameter implants may be associated with greater marginal bone loss. These findings should be regarded cautiously, due to the short follow-up duration and methodological heterogeneity of the primary studies. Further studies should address these aspects in greater detail.

## Supporting information

**S1 PRISMA Checklist.**
(PDF)

## Acknowledgments

The authors thank Professors Carlos Alberto Feldens, Gustavo Frainer Barbosa, and Otacílio Luís Chagas Júnior for their important suggestions during the conduction of this study.

## Author Contributions

**Conceptualization:** Lucas Henrique Telles, Fernando Freitas Portella, Elken Gomes Rivaldo.

**Data curation:** Lucas Henrique Telles, Fernando Freitas Portella, Elken Gomes Rivaldo.

**Formal analysis:** Lucas Henrique Telles, Fernando Freitas Portella, Elken Gomes Rivaldo.

**Funding acquisition:** Lucas Henrique Telles, Fernando Freitas Portella, Elken Gomes Rivaldo.

**Investigation:** Lucas Henrique Telles, Fernando Freitas Portella, Elken Gomes Rivaldo.

**Methodology:** Lucas Henrique Telles, Fernando Freitas Portella, Elken Gomes Rivaldo.

**Project administration:** Lucas Henrique Telles, Fernando Freitas Portella, Elken Gomes Rivaldo.

**Software:** Fernando Freitas Portella.

**Supervision:** Lucas Henrique Telles, Fernando Freitas Portella, Elken Gomes Rivaldo.

**Validation:** Lucas Henrique Telles, Fernando Freitas Portella, Elken Gomes Rivaldo.

**Visualization:** Lucas Henrique Telles, Fernando Freitas Portella, Elken Gomes Rivaldo.

**Writing – original draft:** Lucas Henrique Telles, Fernando Freitas Portella, Elken Gomes Rivaldo.

**Writing – review & editing:** Lucas Henrique Telles, Fernando Freitas Portella, Elken Gomes Rivaldo.

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
