## [Decision Letter · Decision Letter 0]

26 Sep 2019

PONE-D-19-22348

Longevity and marginal bone loss of narrow-diameter implants supporting single crowns: a systematic review

PLOS ONE

Dear Prof. Portella,

Thank you for submitting your manuscript to PLOS ONE. After careful consideration, we feel that it has merit but does not fully meet PLOS ONE’s publication criteria as it currently stands. Therefore, we invite you to submit a revised version of the manuscript that addresses the points raised during the review process.

The reviewers and editor in general were please with the manuscript. However, there is an important issue of lack of included studies. It is therefore important to carefully discuss the papers excluded that might have some insight into the topics. Please include these important excluded articles in the Discussion. Also, please expand the limitations and how you might overcome this in the future.

We would appreciate receiving your revised manuscript by Nov 10 2019 11:59PM. To enhance the reproducibility of your results, we recommend that if applicable you deposit your laboratory protocols in protocols.io, where a protocol can be assigned its own identifier (DOI) such that it can be cited independently in the future. For instructions see: http://journals.plos.org/plosone/s/submission-guidelines#loc-laboratory-protocols

We look forward to receiving your revised manuscript.

Kind regards,

Sompop Bencharit, DDS, MS, PhD, FACP

Academic Editor

PLOS ONE

Journal Requirements:

Please provide an amended Funding Statement that declares *all* the funding or sources of support received during this specific study (whether external or internal to your organization) as detailed online in our guide for authors at http://journals.plos.org/plosone/s/submit-now.  Please state what role the funders took in the study.  If any authors received a salary from any of your funders, please state which authors and which funder. If the funders had no role, please state: "The funders had no role in study design, data collection and analysis, decision to publish, or preparation of the manuscript."

Reviewers' comments:

Reviewer's Responses to Questions

**Comments to the Author**

1. Is the manuscript technically sound, and do the data support the conclusions?

Reviewer #1: Yes

Reviewer #2: Yes

2. Has the statistical analysis been performed appropriately and rigorously? 

Reviewer #1: Yes

Reviewer #2: Yes

3. Have the authors made all data underlying the findings in their manuscript fully available?

Reviewer #1: Yes

Reviewer #2: Yes

4. Is the manuscript presented in an intelligible fashion and written in standard English?

Reviewer #1: Yes

Reviewer #2: Yes

5. Review Comments to the Author

Reviewer #1: In this systematic review with meta-analysis, Telles et al. compared marginal bone loss and longevity around narrow- and standard-diameter implants supporting single crowns. Four studies were included in the quantitative review. Authors observed no significant difference between groups for longevity. They observed more marginal bone loss around narrow implants after up to 3 years follow-up. Nevertheless, there was no significant difference when limiting the analyses to RCTs only.

This is a nice paper dealing with a topic of main clinical importance. Few studies were included in the review, due to paucity of available data. Please find below my comments :

Introduction :

Last sentence : please (1) change ‘the present study’ with ‘the present systematic review’ and (2) add ‘supporting single crowns’, which is the strenth of the present paper.

Methods :

Authors declared propspective studies were excluded, but Santing et al. was declared in the manuscript as a prospective study (p8). Also, looking at references, Santing et al. study looks to be Hendrick et al. study.

Authors included scientific literature recommended by implant manufacturers, which exposed them to bias of sponsorship. Also, did the authors perform an electronic seach in EMBASE ?

Authors should use the Kappa statistic to assess interrater reliability between reviewers.

Results :

Line 190 : Andersen is a clinical trial, not a randomized clinical trial.

What about the fifth study (Gi et al.) ? It is a duplicate with Ionnidis et al, but because this study was indicated in the qualitative analysis (Figure 1), 1-year follow-up data should be indicated in both manuscript and Table 1.

Please have a look on success rates for Santing et al. and De Souza et al. studies. Data indicated in the manuscript differ from those indicated in Table 1.

Discussion :

First sentence : please indicate that the review included only studies with implants supporting single crowns (this is a strength).

Line 240 : Unless I misunderstood, for a same initial bone volume, bone tissue surrounding a narrow diameter implant is greater around a standard diameter implant, because the implant diameter is lower.

Among the three studies which compared marginal bone loss around narrow- and standard-diameter implants, one study used immediate provisionalization (Andersen et al.), whereas the others did not. This immediate functionnal loading is associated with occlusal forces which are directed along the long axis of the implant body. The resulting contact pressure at the implant-bone interface may interfere with osseointegration. Because the lower the surface is, the higher the pressure is, authors may also hypothesize that higher contact pressure at the implant-bone interface for immediate provisionalized narrow diameter implants results in higher marginal bone loss.

Tables and figures :

Figure 1 : Please have a look on the number of ‘records identified through database searching’ and the number of ‘additional records identified through hand search of reference lists’. When looking to the number of duplicates, it suppose that nearly all studies were identified through the four sources (Pubmed, Scopus, Scielo and hand search), which is very very surprising.

Table 1 : Please indicate references numbers. For Santing et al (2012) study, Table 1 indicates 10 narrow diameter implants whereas manuscript indicates 12 narrow diameter implants.

Reviewer #2: This manuscript reviews the success and bone loss around narrow diameter implants.

The introduction is well structured and prevents the case for this review well.

Methods are appropriate and well documented. What is called "success" is actually survival. Some assessment of clinical parameters would be good or using previous criteria such as Karoussis 2003/2004.

It's not a surprise only 4 papers could be included.

Since the authors make the comments about the implant material, I feel this should be included in Table 1.

Is it possible to undertake more analysis of the where the implants were placed. I would think it would be more anteriorly. What was the effect of position on the bone loss? Do posterior narrow implants have more bone loss?

Is there any information on why the natural teeth were lost? Was it Perio, caries, trauma, etc. If it's Perio then this may account for the higher bone loss?

Discussion is appropriate for the results.

References appropriate and up to date.

6. PLOS authors have the option to publish the peer review history of their article (what does this mean?). If published, this will include your full peer review and any attached files.

Reviewer #1: No

Reviewer #2: No

---

## [Author Response · Author response to Decision Letter 0]

23 Oct 2019

Dear Professor Sompop Bencharit

On behalf of my coauthors, I would like to thank you for the time spent in reviewing our manuscript “Longevity and marginal bone loss of narrow-diameter implants supporting single crowns: a systematic review” (PONE-D-19-22348).

We reiterate our wish to publish in PLOS ONE. Please find a point-by-point response to the reviewers’ comments below, in italic. Changes to the manuscript are highlighted in yellow.

Reviewer #1

In this systematic review with meta-analysis, Telles et al. compared marginal bone loss and longevity around narrow- and standard-diameter implants supporting single crowns. Four studies were included in the quantitative review. Authors observed no significant difference between groups for longevity. They observed more marginal bone loss around narrow implants after up to 3 years follow-up. Nevertheless, there was no significant difference when limiting the analyses to RCTs only.

This is a nice paper dealing with a topic of main clinical importance. Few studies were included in the review, due to paucity of available data. Please find below my comments :

Introduction :

Last sentence : please (1) change ‘the present study’ with ‘the present systematic review’ and (2) add ‘supporting single crowns’, which is the strenth of the present paper.

We thank the reviewer for this assessment of our research, and for the time devoted to reviewing our manuscript. We believe these comments and suggestions substantially improved the quality of our work.

Methods :

Authors declared propspective studies were excluded, but Santing et al. was declared in the manuscript as a prospective study (p8). Also, looking at references, Santing et al. study looks to be Hendrick et al. study.

Thank you for this observation. Indeed, prospective observational studies were excluded. Santing et al. presented prospective data of an interventional study. Santing et al. and Hendrick et al. represent the same article; this was an error made during reference formatting. We apologize, and have double-checked all references. 

Authors included scientific literature recommended by implant manufacturers, which exposed them to bias of sponsorship. Also, did the authors perform an electronic seach in EMBASE ?

We agree that selection bias can be an issue when citing literature recommended by the implant industry. However, all of the included articles retrieved from these lists were also found by our electronic database search. For the sake of transparency, a sentence stating that none of the included articles was retrieved solely from manufacturers’ lists has been added to the manuscript. We also considered sponsorship bias in the Risk of bias analysis.

EMBASE was not searched. 

Authors should use the Kappa statistic to assess interrater reliability between reviewers.

The concern regard reliability between reviewers is important. We did not calculate the Kappa statistic because:

- For an article to be excluded during the title/abstract review stage, both reviewers were required to classify the paper as ineligible; if they disagreed, the paper was read in full;

- Full-text reading: evaluation for eligibility and data extraction was performed together. During this process, no article gave rise to any debate or disagreement.

Results :

Line 190 : Andersen is a clinical trial, not a randomized clinical trial.

Thank you for noting. We have corrected this sentence.

What about the fifth study (Gi et al.) ? It is a duplicate with Ionnidis et al, but because this study was indicated in the qualitative analysis (Figure 1), 1-year follow-up data should be indicated in both manuscript and Table 1.

Information regarding 1-year follow-up has been added to the manuscript.

Please have a look on success rates for Santing et al. and De Souza et al. studies. Data indicated in the manuscript differ from those indicated in Table 1.

We apologize for this error. The values in Table 1 have been corrected.

Discussion :

First sentence : please indicate that the review included only studies with implants supporting single crowns (this is a strength).

Grateful for the tip. The sentence has been rephrased.

Line 240 : Unless I misunderstood, for a same initial bone volume, bone tissue surrounding a narrow diameter implant is greater around a standard diameter implant, because the implant diameter is lower.

We agree with the reviewer. This paragraph has been reviewed.

Among the three studies which compared marginal bone loss around narrow- and standard-diameter implants, one study used immediate provisionalization (Andersen et al.), whereas the others did not. This immediate functionnal loading is associated with occlusal forces which are directed along the long axis of the implant body. The resulting contact pressure at the implant-bone interface may interfere with osseointegration. Because the lower the surface is, the higher the pressure is, authors may also hypothesize that higher contact pressure at the implant-bone interface for immediate provisionalized narrow diameter implants results in higher marginal bone loss.

Thank you for this observation. We have added a paragraph to discussion to address this issue.

Tables and figures :

Figure 1 : Please have a look on the number of ‘records identified through database searching’ and the number of ‘additional records identified through hand search of reference lists’. When looking to the number of duplicates, it suppose that nearly all studies were identified through the four sources (Pubmed, Scopus, Scielo and hand search), which is very very surprising.

This is a valid point. Indeed, almost of the papers identified through hand searching were also found in the electronic databases.

Table 1 : Please indicate references numbers. For Santing et al (2012) study, Table 1 indicates 10 narrow diameter implants whereas manuscript indicates 12 narrow diameter implants.

Grateful for the observation. Reference numbers are now cited in the Table, and the number of narrow implants in the Santing study has been corrected.

Reviewer #2:

This manuscript reviews the success and bone loss around narrow diameter implants.

The introduction is well structured and prevents the case for this review well.

Methods are appropriate and well documented. What is called "success" is actually survival. Some assessment of clinical parameters would be good or using previous criteria such as Karoussis 2003/2004.

We agree with the reviewer regarding the definition of implant outcomes. However, due to the short follow-up of primary studies (up to 3 years), we chose to term the analysed outcome “success”, not survival, and stated in the Methods section the definition adopted: “Implants were considered successful when they were osseointegrated and remained functional”. 

It's not a surprise only 4 papers could be included.

Since the authors make the comments about the implant material, I feel this should be included in Table 1.

Thank you for your assessment. This is important information and has been added to Table 1.

Is it possible to undertake more analysis of the where the implants were placed. I would think it would be more anteriorly. What was the effect of position on the bone loss? Do posterior narrow implants have more bone loss?

The region of implant placement is extremely relevant. Unfortunately, variability across studies precluded a more in-depth analysis. We have discussed this issue in greater depth now.

Is there any information on why the natural teeth were lost? Was it Perio, caries, trauma, etc. If it's Perio then this may account for the higher bone loss?

Information and discussion regarding this issue was considered during manuscript review. A paragraph to this effect has been added.

Discussion is appropriate for the results.

References appropriate and up to date.

Once again, we would like to thank the Editor and the Reviewers for the important suggestions and constructive criticism. We believe the quality of the paper has improved significantly as a result. Thank you for your consideration.

Sincerely,

Fernando Freitas Portella (on behalf of all authors)

---

## [Editor Report · Decision Letter 1]

29 Oct 2019

Longevity and marginal bone loss of narrow-diameter implants supporting single crowns: a systematic review

PONE-D-19-22348R1

Dear Dr. Portella,

We are pleased to inform you that your manuscript has been judged scientifically suitable for publication and will be formally accepted for publication once it complies with all outstanding technical requirements.

With kind regards,

Sompop Bencharit, DDS, MS, PhD, FACP

Academic Editor

PLOS ONE

Additional Editor Comments (optional):

The authors had sufficiently addressed all comments from the reviewers. Thank you for the revision.
---

## [Editor Report · Acceptance letter]

1 Nov 2019

PONE-D-19-22348R1 

Longevity and marginal bone loss of narrow-diameter implants supporting single crowns: a systematic review 

Dear Dr. Portella:

I am pleased to inform you that your manuscript has been deemed suitable for publication in PLOS ONE. Congratulations! Your manuscript is now with our production department. 

With kind regards,

on behalf of

Dr. Sompop Bencharit 

Academic Editor

PLOS ONE